# Towards a Better Use of Antimicrobials on Farms: Insights from a Participatory Approach in the French Pig and Poultry Sectors

**DOI:** 10.3390/antibiotics11101370

**Published:** 2022-10-07

**Authors:** Christian Ducrot, Marie-Jeanne Guénin, Anne Hemonic, Nathalie Rousset, Yannick Carre, Charles Facon, Philippe Le Coz, Jocelyn Marguerie, Jean-Marc Petiot, Maxime Jarnoux, Mily Leblanc-Maridor, Mathilde Paul, Sophie Molia, Catherine Belloc

**Affiliations:** 1ASTRE, Université de Montpellier, CIRAD, INRAE, 34398 Montpellier, France; 2IFIP-Institut Du Porc, La Motte au Vicomte, 35651 Le Rheu, France; 3ITAVI, Antenne Ouest, 22440 Ploufragan, France; 4ANVOL, Interprofession Volaille de Chair, 175 Rue Jean Monnet, 29490 Guipavas, France; 5SNVECO, Syndicat National des Vétérinaires COnseils, 23, Rue Olivier de Serres, 85500 Les Herbiers, France; 6Société Nationale des Groupements Techniques Vétérinaires, 75011 Paris, France; 7Conseil National de l’Ordre des Vétérinaires, 34 Rue Bréguet, 75011 Paris, France; 8DGAL, Direction Générale de L’alimentation, 251 Rue de Vaugirard, 75015 Paris, France; 9BIOEPAR, INRAE, Oniris, 44300 Nantes, France; 10IHAP, Université de Toulouse, INRAE, ENVT, 31076 Toulouse, France

**Keywords:** antimicrobial use, pig, poultry, health, welfare, participatory approach, monitoring tools, drug managing, antibiotic-free meat

## Abstract

Despite the strong decrease in antimicrobial use in the French poultry and pig sectors over the last decade, room for improvement remains. A participatory approach was set up in France, involving representatives of veterinarians, the pig and poultry industries, technical institutes, the French Ministry of Agriculture, and researchers, to further improve how antimicrobials are used on farms. By successively defining a shared, long-term vision of future antimicrobial use on farms, identifying lock-in mechanisms impeding this future vision from being realized, and articulating practical questions on how to move in the desired direction, the group rapidly reached a consensus. The results highlight the need for consensual standardized monitoring tools that would allow farmers and veterinarians to jointly monitor the health, welfare, antimicrobial resistance, and antimicrobial use on farms. Other results relate to better communication and training for citizens regarding animal health, animal welfare, and proper antimicrobial use; some benefits but also counterproductive effects of antibiotic-free labels that imperil animal health and welfare; the economic competitiveness of farms on international markets; and the economic sustainability of farm animal veterinary practices. These results call for a concerted way to produce tools for farmers and veterinarians and the broader involvement of other food sector actors.

## 1. Introduction

The overuse and improper use of antimicrobials on humans and animals worldwide has contributed to the emergence and spread of antimicrobial resistance (AMR) [1]. Within the last 15 years, and thanks to various initiatives at the international, European, and French levels [2,3,4], much has been accomplished in terms of reducing the use of antimicrobials on farm animals. At the European level, their use as growth promotors has been forbidden since 2006 [5], and their use in prevention and treatment has decreased drastically. In France, the average amount of antimicrobial use per pig decreased by 56% and that for poultry decreased by 64% over the past ten years [6]. Previous research has shown that antimicrobial use is highly dependent on humans’ perception of disease and risk [7,8]. Antimicrobial treatment is not a clear-cut decision based only on rational indicators, and the prescription of antimicrobials involves pressure, negotiation, and compromise between veterinarians, farmers, production organizations, and brand owners [9,10]. Changing attitudes toward antimicrobial use can be difficult due to lock-in mechanisms and the social environment [9]. Previous findings have called for multiactor initiatives to reduce antimicrobial use, expanding the focus beyond veterinarians and farmers to include downstream operators and policymakers [9]. However, the diverging interests and visions of antimicrobial use held by these various stakeholders render it difficult to achieve a shared objective. Interdisciplinary approaches based on social science frameworks have proven useful in addressing challenges associated with behavior change in the animal health sector, including antimicrobial use [11]. Participatory approaches offer a promising perspective for integrated approaches to health [12]. Among them, living labs allow solutions to be tested in contextual situations that are co-constructed by including knowledge jointly produced by local actors and researchers [13].

In this context, a multiactor participatory approach including veterinary practitioners, technical institutes, interprofessional organizations, public authorities, and researchers was implemented in France to reach a collective vision of long-term objectives and the obstacles that need to be lifted to further improve the use of antimicrobials in the poultry and pig sectors. This approach was used to establish the foundations of a living lab and identify issues that could then be addressed through an intervention. The purpose of the present article is to present the shared vision of the group and its analysis of the lock-in mechanisms and obstacles that must be removed to improve the use of antimicrobials in the French poultry and pig sectors.

## 2. Results

The participatory approach used in the project, detailed by Guenin et al. [14], allowed the participants to move collectively through a succession of steps. They started with an initial diagnosis of the situation then moved on to define a shared vision of a desired middle-term future concerning antimicrobial use in the pig and poultry sectors. The process ended with a detailed analysis of the obstacles that need to be surmounted to reach that desired vision.

### 2.1. Vision of the Future for the Use of Antimicrobials on Farms

The initial diagnosis shows that the implemented efforts and initiatives have led to a significant reduction in antimicrobial use in the pig and poultry sectors, but a threshold has been reached below which it is difficult to achieve further quantitative reductions. The group also agreed that many indicators exist that can be used to monitor the evolution of antimicrobial use, but these indicators are not always used in an optimal manner. The group also noted that although “antibiotic-free” charters have contributed to reductions in use, they can pose challenges for farmers, as antimicrobial treatments are sometimes necessary to ensure animal health and welfare. Reflecting on the initial diagnosis, the group of actors agreed on a vision of the future. Consensus was reached quickly and was worded in the following way:


*“In 2031, in France, the proper use of antimicrobials in the poultry and pork industries focuses on “better” and not just “less” antimicrobial use, this is applied in all farms and accepted by actors involved in the use of antimicrobials (veterinarians, farmers, production organizations, pharmaceutical industries, purchasing centers, etc.) and by those involved in the use of animal products (slaughterhouses, retail, catering, consumers, etc.). Antimicrobial use, monitored by appropriate indicators, makes it possible to preserve the therapeutic arsenal while guaranteeing, on the one hand, the health and welfare of farm animals, and on the other, the economic sustainability of the pig and poultry industries and those of the veterinary network in the country.”*


### 2.2. Central Issue

To move toward this vision of the future, the main reasons that this vision has not yet been reached had to be identified to define the scope of the intervention. The group focused its attention on two key areas, formulating the central issue as follows:


*“Pork and poultry consumption choices by consumers do not systematically take into account the use of antimicrobials. Meanwhile, field actors (veterinarians, farmers, production organizations, etc.) lack data or make heterogeneous use of monitoring indicators of health, welfare and antimicrobial use on animals, which should allow them to tailor their practices in terms of treatment choices and farm management.”*


### 2.3. Problem Tree

All the reasons agreed on by the group that contributed to the current situation stated in the central issue were then listed and organized by the group members in a detailed problem tree [14]. After analysis, reorganization, and the combination of related ideas to simplify their presentation, the global branches of the problem tree were laid out. The tree is presented in Figure 1, and the precise points of each branch are detailed in Table 1.

## 3. Discussion

The participatory approach used in this work showed that different pig and poultry sector stakeholders could agree on a shared long-term goal and vision of the future concerning the use of antimicrobials, despite distinct and sometimes competing professional interests. In the present work, consensus was reached quite rapidly with the help of the ImpresS ex ante method. This method, in effect, starts with a long-term vision that is easier for participants to reach, and then focuses on barriers without stigmatizing particular actors. Farming activities also have been put on the spot by different elements of society regarding animal welfare, the safety of animal products, their contribution to global warming, and difficulties in feeding the world population [15]. In this context, and for the viability and durability of the industry, animal farming needs to be conducted flawlessly, particularly regarding the use of antimicrobials and welfare issues; this puts pressure on the actors to find consensual solutions.

It is quite clear that all of the participants agreed that the better use of antimicrobials needs to be promoted while ensuring animal health and welfare. In France, antibiotic-free charters have been developed in the last ten years. These labels, organized by private operators from the poultry and pig sectors, are based on quantitative objectives (the percentage of flocks treated with antimicrobials) and result in financial bonuses or maluses for the farmers, which vary depending on antimicrobial treatments. All of the actors emphasized that the development of these labels has led to a situation where some farmers prefer not to treat sick animals (or accept mortality, for example, in poultry) rather than lose the financial bonus. This situation, which harms animal welfare, should be ended without delay. The group therefore insisted on the need to monitor precisely, at the farm level, not only antimicrobial use but also health and welfare indicators, so as to be able to document the trade-offs and synergies that exist between antimicrobial use, animal health, and welfare. After a decade of virtuous effects and a strong reduction in antimicrobial use, antibiotic-free labels should also now be revised to continue promoting the limited use of antimicrobials while ensuring constant respect for animal welfare.

The participatory method used in this work also made it possible to share diverse knowledge, experiences, and points of view among the participants and therefore to deeply explore the various factors that explain the current unsatisfying situation regarding antimicrobial use. Given the limited time available, and to make facilitation and participation during workshops easier, we did not involve consumer or citizen representatives. The group also included only one person from each interbranch organization who represented the various actors of the industry, from upstream (farmers) to downstream (retailers). If the veterinarian practitioners were pretty well represented from different perspectives (a technical organization, the union of veterinarians, and the national Council of Veterinary Surgeons), the different components of the production chain were less represented. Enlarging the panel would have given us a broader view of the situation and complementary positions. Moreover, there are various competing firms in the field that may not share the same views on the situation and would not feel well-represented by the interbranch organization. A further step could be to present and discuss results from this work with other actors from the industry and society to validate future interventions that could be implemented in the field. The size of the group proved to be appropriate for being able to handle the participatory approach that is well-designed to facilitate the involvement of the participants.

One of the main points that arose during the group discussions concerned indicators and monitoring tools at the farm level. Various private tools have been tested in France to monitor the use of antimicrobials on farms. These tools are based on data collected from veterinarians’ prescriptions, but the indicators are not calculated in the same way, and they are not widely used. Furthermore, these private initiatives are implemented by different animal production companies as market differentiators and thus are not shared between farms and operators. In order to overcome these lock-in mechanisms, there is a strong need to use standardized tools to monitor antimicrobial use, either based on veterinarians’ prescriptions (with the future CALYPSO project, which will be the French tool to apply EU regulation 2019/6 of 11 December 2018 to veterinary medicinal products) and/or from data directly recorded by farmers in their registers of treatment (for example, the GVET tool in the pig sector). Furthermore, these tools also need to incorporate precise monitoring of health and welfare to promote a global approach to health and welfare monitoring. Strong support from public authorities is needed to tackle this complicated problem and to reach an agreement on the objectives, appropriate indicators, pre-existing tools to be reused (if existing and adapted to produce the indicators), and interconnection of existing databases. Further research projects involving stakeholders are also required to develop and test new tools using both applied and research perspectives to analyze in detail the synergies and trade-offs that exist between health, welfare, and antimicrobial use at different scales (animal, flock, and farm). The monitoring of antimicrobial use at the farm level has already been implemented for years in different species and countries (a review in [16]). In Denmark, for example, this monitoring is used for the green card system in the pig sector [17]. If the cultural context is different and the device cannot be extrapolated, the principles of monitoring antimicrobial use at the farm level, combined with a tailor-made action plan established with the support of veterinarians, could successfully reduce antimicrobial use without jeopardizing production parameters [18]. Regarding farm monitoring, the participants in the present work also highlighted the need to measure and monitor antimicrobial resistance at the farm level to highlight the potential relationship with antimicrobial use on the farm and to guide farmers’ and veterinarians’ prescriptions. This would require applied research to adapt existing surveillance tools and procedures at the farm level; design monitoring devices targeted to the commensal flora, manure, and the environment; test their accuracy in terms of sensitivity and specificity; and work on cheaper options so that such surveillance is affordable.

Many of the lock-in mechanisms identified by the group relate to farmers’ practices, their willingness to invest in prevention, and their financial capacity to invest in new equipment and facilities to improve the hygiene and indoor environment (temperature, hygrometry, and ventilation) of livestock buildings. Previous studies have reported the influence of high production costs and an inability to invest as factors that influence antimicrobial usage on farms and the avoidance of investments in prevention [9]. Furthermore, studies have shown that the risk of resistance to antimicrobials is not a concern for many farmers [9]; their decision to move to better prevention and less use of antimicrobials is usually not driven by a fear of antimicrobial resistance or public health issues but rather by other considerations such as sanitary problems, economic bonuses, a change in manager (for example from father to son), or a move to organic farming [19]. It is therefore difficult for advisors, including veterinarians, to better manage antimicrobial use by farmers who do not recognize the problem, want to invest, or accept changes that would affect their daily routines.

It should also be mentioned that, in many French pig and poultry farms, antimicrobial use has been decreased to a minimum and cannot be reduced further without compromising animal health and welfare. To take action on these subjects at the farm level, previous studies have shown the important role of production organizations [20]. In poultry and pig production, most farmers contract with a production organization to sell their animals to the market. These organizations have a strong economic base to finance and support different actions involving farmers (technical support, training, and economic bonuses in antibiotic-free supply schemes), and they have the power to impose actions on farmers. These organizations are themselves under constant pressure from brands, retailers, and consumers to further reduce antimicrobial use on farms. In order to facilitate these transitions, other aspects should also be investigated. Restrictive regulations regarding antimicrobial prescriptions (the choice of antimicrobial, dosage, and duration of treatment) have been noted as factors limiting the smarter use of antimicrobials. While these regulations are difficult to change because they were implemented to protect animal and public health, they could be updated in the light of new scientific evidence. Currently, veterinarians base an antimicrobial prescription on a systematic bacteriological examination followed by an antibiogram. Practitioners postulate that new technical levers of action may exist that we need to discover, reinforcing the interest in continuing applied investigations and practical work based on farm experiments.

Another series of lock-in mechanisms concern veterinary practitioners. In France, veterinarians have considerably modified their antimicrobial use practices over the last 15 years, with specialists on the subject meeting to reach a consensus on recommended technical actions [21]. However, the technical evolution of their practice put them in a risky economic situation because their economic model remains largely based on the sale of drugs [22,23]. They sell fewer and fewer antimicrobials, and the products that could partly replace this loss of revenue, such as nutrition and phytotherapy products, are also sold by other supply companies because they are unregulated. The technical expertise that veterinarians provide farmers should be valued, but until now most farmers and production organizations have been unwilling to pay for this advice. This situation is economically untenable for the profession, and there is a strong risk that the network of veterinarian practitioners in rural areas will be disrupted. This would be detrimental for veterinary public health issues, for example, the surveillance of antimicrobial resistance and the prescription of antimicrobials as well as the detection and management of zoonoses and other economically devastating animal diseases (including African swine fever and avian influenza). Young veterinarians quickly lose their motivation to work on farm animals, partly due to the nonrecognition of their work and the uncertainties of the sustainability of veterinary practices. Apart from this major socioeconomic issue, some expectations voiced by the veterinary profession relate to practical tools that would allow them to better diagnose and prescribe, such as diagnosis tools, antibiograms on farms, and alternative therapeutics. Developing such products requires manufacturers and drug companies to be convinced of the economic potential and return on investments in such products.

Other results of the participatory approach involve questions that extend beyond the practical use of antimicrobials on farms and involve other actors, such as retailers and consumers. Retailers designed antibiotic-free labels in recent years because there is a market for such products, which can be sold at higher prices. Few studies analyzed in detail consumers’ understanding and knowledge of antimicrobial resistance. A scoping review about consumer perceptions of antimicrobial use in animal husbandry confirms a willingness to pay a premium for antibiotic-free products that varies strongly depending on the investigated geographic, social, and cultural settings [24]. More interestingly, while trying to untangle the answers provided by consumers, the authors found that the major threats regarding antimicrobial use in husbandry perceived by consumers include consumer safety because they fear the presence of antimicrobial residues in animal products as well as the association of antimicrobial use with poor animal welfare, with the concern about antimicrobial resistance being less pronounced. The use of antimicrobials on farms echoes the various grievances that consumers have against animal farming that are detailed in [25], including the impact of farming activities on the environment and climate, the welfare of animals, the intensification of farming activities, and health issues. In the same vein, another study evidenced concerns regarding cramped conditions leading to higher disease prevalence and the prophylactic use and overuse of antimicrobials [26]. As stated by the participatory group, and from a societal perspective, the question of antimicrobial use is mixed with other negative images about animal farming and cannot be dissociated in the minds of consumers or treated separately. Strong actions leading to more acceptable practices regarding animal health and welfare and a reduced impact of farming on the environment should help to improve the image of animal farming. In the minds of consumers, antibiotic-free labels are probably considered as indicators of better farming conditions that render the use of antimicrobials unnecessary. In reality, however, the situation on farms is likely more complex. The better use of antimicrobials does not mean the absence of antimicrobials because in some instances, even when farming practices are improved, the use of antimicrobials is a necessity. Another point is the need for clear and harmonized antibiotic-free labeling that would help to inform consumers instead of confusing them.

The participatory approach presented in this paper took place in a specific context, namely the poultry and pig sectors in France. Both industries share various aspects concerning farming (batch of animals), veterinary practice (herd medicine), and the use of antimicrobials (with a huge decrease in the last 10 years in France), so they could be processed together in the participatory approach. The question therefore arises as to how much the results can be extrapolated and generalized. Depending on the country, mentalities and developments are more or less advanced concerning the reduced use of antimicrobials on farms; however, different papers tend to show that some results are shared. Hockenhull et al. [9] reported, in different studies, the lack of knowledge on the part of farmers concerning the risks involved in the use of antimicrobials, their reluctance to increase production costs and invest in prevention to reduce the need for antimicrobials, their pressure on veterinarians for antimicrobial prescriptions, and that barriers to change are largely shared. The monitoring of antimicrobial use has been tested in different studies, including making it part of a regulation device in certain countries [17]. Concerning consumers, the misunderstanding of antimicrobial resistance versus antimicrobial residues in animal products is also observed in other countries [24,27]. All of these points lead us to believe that our results may be relevant to other contexts.

## 4. Materials and Methods

In order to initiate a living lab in the French pig and poultry sectors and identify issues that could then be addressed through an intervention, the participatory approach called ImpresS ex ante was implemented [28], inspired by outcome mapping and program theory [29,30].

### 4.1. Principle of the Participatory Approach Used

The ImpresS ex ante approach was designed to improve the impact of research and development programs. The main idea of this framework is to formulate a common vision of the target and a plausible impact pathway of an intervention and then to identify sticking points to be dealt with during the process. This methodological support includes a systemic analysis of the intervention context. By taking into account the context and the expectations formulated by the actors, the ImpresS ex ante method thus aims to improve the plausibility of the codesigned strategies to generate the necessary changes and the desired long-term impacts. The steps and the way this approach is conducted follow specific rules that are detailed in Guenin et al. [14].

### 4.2. Actors Involved

To increase the plausibility of the formulated statements, the group was set up to bring together representatives of the stakeholders involved or concerned by the use of antimicrobials in farm animals who could contribute expertise and a particular perspective on the context. Each participant represented an organization that could play a role in the process of innovation regarding the use of antimicrobials and in the construction of strategies.

Nine participants were mobilized throughout the participatory process: two veterinary practitioners involved in the issue of antimicrobial resistance, specializing in the pig or poultry sectors, and representing the National Veterinary Technical Society; two veterinarians representing the Chamber of Veterinary Surgeons and the National Union of Veterinary Advisors; an engineer and a veterinarian from the technical institutes for pork and poultry farming; two engineers from the interbranch organizations of the pig and poultry industry, representing the different professional categories of the industry (one of them attended only one meeting); and a member of the Ministry of Agriculture in charge of antimicrobial resistance issues. The design and facilitation of the participatory process was conducted by a research team that included an expert on antimicrobial issues and pig and poultry farming (C.B.) and a person trained in participatory approaches and the ImpresS ex ante method (M.J.G.). A third researcher was in charge of checking the correct application of this method (S.M.). Four researchers also observed the participatory workshops to later analyze the process from a sociological perspective.

### 4.3. Implementation of the Method

The participatory process reported in this article took place during four full-day meetings that were held over a nine-month period from May 2021 to February 2022. Because of the COVID-19 pandemic, one of the meetings was conducted entirely by videoconference. The other three meetings were conducted on site with the possibility for some participants to participate online. For each meeting, specific objectives and expected outputs were defined by the research team. The successive stages, which were spread out over the four meetings, consisted of drawing up an initial diagnosis of the current situation with regard to the use of antimicrobials (narrative) then agreeing on a vision of the desirable future in 10 years (narrative), which the group sought to contribute to through an intervention. The group then defined a central issue that was, from their point of view, the main reason the vision of the future had not yet been reached and that they felt able to help address (narrative). They then shared ideas and agreed on the major reasons explaining this central issue, which were organized in a problem tree (diagram). Further stages were dedicated to deepening the paths of change on certain branches of the problem tree, but they are beyond the scope of this article and are not reported here. The results obtained at each stage of the process (vision of the future, central issue, problem tree, etc.) were submitted in a report to all participants and were subsequently rediscussed to make sure all participants approved them.

The problem tree was developed through a live chat, which meant that various ideas sometimes overlapped or varied in terms of precision. On the basis of this complete diagram, which is presented in Guenin et al. [14], we analyzed, combined, and reorganized ideas in order to simplify the whole picture without altering the main meaning of the ideas expressed by the group.

## 5. Conclusions

The participatory approach carried out with participants from the poultry and pig sectors in France made it possible to frame a shared view of the future and identify the lock-in mechanisms that must be lifted for a better, and not just decreased, use of antimicrobials on farms. This is a very positive report since working on the complex issues that were raised requires a pluridisciplinary and global approach. The ideas issued from the group were probably present in the minds of many veterinarians; the real novelty is that various stakeholders with diverging interests in the production chain were able to agree to prioritize these items as major components to improve the use of antimicrobials in the pig and poultry sectors. Some issues raised by the group concern the responsibilities that they share, such as implementing standardized and widely used monitoring tools on farms to monitor animal health, animal welfare, antimicrobial use, and antimicrobial resistance. Others go beyond their own responsibilities and the specific question of antimicrobials and deal with negative societal images of husbandry practices and the misunderstanding on the part of many consumers regarding antimicrobial resistance and antimicrobial use. Another important point that arose from the work is the critical economic situation and the meaning given to the work of the rural veterinary profession. Strong government support might help to boost initiatives and lift the lock-in effects resulting from economic competition between animal sector operators. The opportunity of the coming update of the French Ecoantibio plan should allow France to push actions on the standardization and spreading of monitoring tools at the farm level and to work on communication with citizens. The participatory approach proved to be a fruitful option to bring stakeholders with divergent interests to discuss a unifying subject.

## Figures and Tables

**Figure 1 antibiotics-11-01370-f001:**
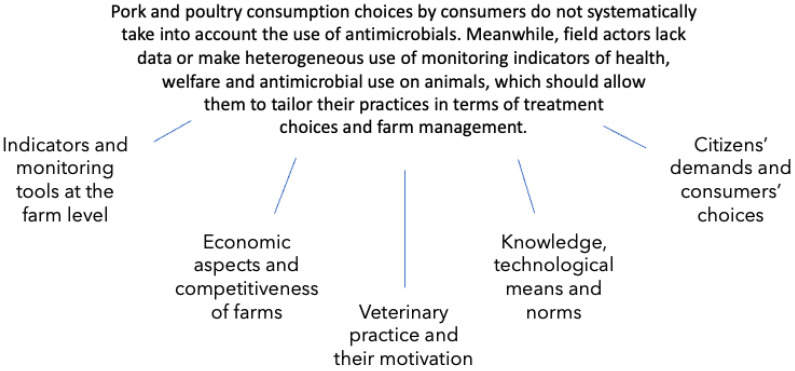
Problem tree displaying the five main categories of lock-in mechanisms that result in the central issue regarding the better use of antimicrobials and prevent the vision of the future from being reached, according to the participants.

**Table 1 antibiotics-11-01370-t001:** Problems raised in the different branches of the problem tree *(explanations)*.

**Problems regarding indicators and monitoring tools at the farm level**
− A lack or misuse of standardized indicators and monitoring tools for antimicrobial use at the farm level; − A lack of monitoring tools at the farm level combining animal health, animal welfare, antimicrobial use, and antimicrobial resistance to better monitor antimicrobial use; − Flows of monitoring data between partners are uneven and suboptimal, impeding better monitoring by the different actors involved.
**Problems regarding economic aspects and the competitiveness of farms**
− Farmers’ perception that preventive measures, including alternatives to antimicrobials (vaccines, biosecurity, hygiene, etc. = fixed costs), are more expensive than curative measures with antimicrobial treatment (variable costs), despite economic studies that prove the contrary; they prefer the variable charge by treating with antimicrobials rather than taking risks on the fixed charge by favoring preventive alternatives; Investing in prevention can affect the competitiveness of farms. Depending on their economic strength, farmers lack the investment capacity to improve farming infrastructure; − Stricter national standards and stricter controls could contribute to degrading the competitiveness of French operators compared to European and international competitors and decrease the acceptability of rules to farmers; − A lack of economic valuation of the achieved results of the reduction in antimicrobial use on farms *(such as incentives or increased price);* − Antibiotic-free specifications generate competition within the poultry sector.
**Problems regarding veterinary practice and their motivation**
− Farmers use unregulated products (such as phytotherapy) as alternatives to antimicrobials. This deteriorates the financing of the veterinary network and services provided to the sectors by switching from prescription care (*income for the veterinarians*) to unregulated free products (*veterinarians are not competitive in this market*); − Antibiotic-free claims (*that discourage farmers from treating sick animals to keep the economic valuation of the charter)* tend to guide antimicrobial use more than health monitoring and care by the veterinarian, which results in a loss of motivation on the part of veterinarians; − The nonseparation of veterinary advice to farmers and drug sales by the veterinarian practitioner could generate conflicts of interest against the reduction in antimicrobial use (listed by some participants because it was debated in the public arena, but it was not a shared opinion); − Veterinarians do not receive incentives for the better use of antimicrobials (*however diligent they are in improving the use of antimicrobials, in contrast to the case of doctors).*
**Problems regarding a lack of knowledge, technological means, and norms**
− A lack of technological means (e.g., robots, apps, and quick antibiograms in pigs) to better target antimicrobial prescription; − A lack of clear evidence of the link between antimicrobial use on farm animals and antimicrobial resistance in humans that would convince actors; − A lack of knowledge on additional obstacles to change and possible technical levers of action that could help innovate on farms to improve the use of antimicrobials; − Restrictions by regulation (marketing authorization) of the alternative usage of antimicrobials (*for example, alternative therapeutic indication or posology*) that could decrease the risk of resistance, a lack of alternatives to antimicrobials, and fewer and fewer therapeutic solutions for minor species in poultry; − Many regulations concerning farming activities (*which tire farmers and do not facilitate their acceptability)* to the detriment of those concerning antimicrobial use; − A lack of consensual antimicrobial use standards defined according to production types that allow their specific technical–economic objectives to be achieved while ensuring animal health and welfare.
**Questions regarding citizens’ demands and consumers’ choices**
**A lack of communication and education on antimicrobial use on farms:** − The multiplicity of antibiotic-free claims generates confusion among consumers (confusion between residues of antimicrobials and resistance to antimicrobials); − The cumulative effect of negative messages about farming conveyed in the media also generates confusion about farming practices, animal health and welfare, the actual nature of antimicrobials and synthetic chemistry, antimicrobial use practices, and food safety and residues in meat, and citizens do not recognize the huge progress made in the use of antimicrobials in farms; − A lack of information on the differences in practices and labeling between countries and a lack of communication on the positive impacts of animal farming also contribute to this confusion and make it difficult to convey positive messages. **Insufficient consumer purchasing power to integrate the antimicrobial use criterion in their consumption choices:** − Consumers voluntarily or involuntarily reduce their food budget.

## Data Availability

The verbatim transcripts of the interviews and work groups cannot be made anonymous, given the small number of participants and their respective positions, so they are not provided.

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
