# Peer review of "Towards a Better Use of Antimicrobials on Farms: Insights from a Participatory Approach in the French Pig and Poultry Sectors"

_antibiotics, 2022, doi:10.3390/antibiotics11101370_

Round 1
Reviewer 1 Report
I commend the authors for a highly participatory approach used in this study. This should be emulated in other countries given the complexity of AMR and the need for a multidisciplinary and multi sectoral approach.
Author Response
We would like to thank the reviewer for his nice comment on the work done.
Reviewer 2 Report
In the manuscript titled "Towards a better use of antimicrobials on farms: insights from a participatory approach in the French pig and poultry sectors" the author tried to explain the existing condition of antimicrobial use in the poultry and pig sector in French and how to better use the antimicrobials to reduce the antimicrobial resistance (AMR). The manuscript is well designed and well written. I have a few minor inquiries or suggestions.
1. What about the title " Towards a better use of antimicrobials in French Pig and Poultry farms: insights from a participatory approach.
2. Author should explain why they selected only poultry and pig farms, why not other livestock farms?
3. Author should explain the existing conditions of laboratory facilities and practices by the veterinarians before prescribing antimicrobials.
4. Author should explain briefly the limitations and gaps in the study
5. Table 1: Veterinarians do not receive incentives for better use of antimicrobials; The sentence should be explained/rephrased.
Author Response
In the manuscript titled "Towards a better use of antimicrobials on farms: insights from a participatory approach in the French pig and poultry sectors" the author tried to explain the existing condition of antimicrobial use in the poultry and pig sector in French and how to better use the antimicrobials to reduce the antimicrobial resistance (AMR). The manuscript is well designed and well written. I have a few minor inquiries or suggestions.
- What about the title " Towards a better use of antimicrobials in French Pig and Poultry farms: insights from a participatory approach.
Thank you for the suggestion. We prefer to keep the initial title because it increases the interest of the paper compared to the proposed title.
- Author should explain why they selected only poultry and pig farms, why not other livestock farms?
We introduced on Line 300 : “Both industries share various aspects concerning farming (batch of animals raised together), veterinary practice (herd medicine opposed mostly to individual medicine for cattle), and use of antimicrobials (huge decrease in the last 10 years in France), so that they could be processed together in the participatory approach”.
Including other productions such as cattle would have included too many people around the table, preventing the process of the participatory approach.
- Author should explain the existing conditions of laboratory facilities and practices by the veterinarians before prescribing antimicrobials.
Veterinarians base their antimicrobial prescription on a systematic bacteriological examination, followed by an antibiogramme. The identification of bacteria is sometimes done with the help of mass spectrometry (especially MALDI-TOF-MS). Furthermore, the veterinary prescription of critically important antimicrobials requires, by regulation, an antibiogramme carried out in a veterinary laboratory.
We added on Line 240 : “Currently, the veterinarians base the antimicrobial prescription on a systematic bacteriological examination followed by an antibiogramme”.
- Author should explain briefly the limitations and gaps in the study
We already presented different limitations of our study in the initial paper, that are the following (initially line 164) : “Given the limited time available, and to make facilitation and participation during workshops easier, we did not involve consumer or citizen representatives. The group also included only one person for each inter-branch organization who represented the various actors of the industry, from upstream (farmers) to downstream (retailers). Enlarging the panel would have given us a broader view of the situation and complementary positions. »
Following the recommendation of the reviewer, we added on line 167 that “if the veterinarian practitioners were pretty well represented from different perspectives (technical organization, union of veterinarians, national Council of Veterinary Surgeons), the different components of the production chain were less represented”. And further in the paragraph (Line 172) : “Also, there are various competing firms in the field that may not share the same views on the situation and would not feel well represented by the inter-branch organization.
- Table 1: Veterinarians do not receive incentives for better use of antimicrobials; The sentence should be explained/rephrased.
In the filed of Human medicine, the French doctors receive incentives for their prudent prescription of antimicrobials, that is not the case for veterinarians. We developed the idea in the paper, in Table 1. : “ Veterinarians do not receive incentives for better use of antimicrobials, (however diligent they are in improving the use of antimicrobials and in contrast wit
Reviewer 3 Report
The result dedicated in the abstract is very general, for example, what tools can “standardized monitoring tools” be? Any recommendation?
The conclusion part doesn’t reflect the result. It is only a suggestion. In the end, what solution was decided? Furthermore, please note that the complexity of sentences is high and long sentences will tire the reader (in the abstract).
Obviously, most of the stated results have already existed in the minds of many veterinarians and other parts of the related industries. State the main solution for coordination between different organizations and teams. Novelty is not well presented.
The keyword is good but not well. The items can improve using Mesh term. My recommendation: Please delete pig; poultry. Add “drug managing”; and “monitoring tools”; “antibiotic-free meat”.
Related to the following sentence, it would be ideal for the authors to give some suggestions for Diagnostic, Control, and Monitoring tools to help future industries and veterinarians distinguish the best antibiotics: “In 2031, in France, the proper use of antimicrobials in the poultry and pork industries focuses on "better" and not just "less" antimicrobial use”.
Here, please give abbreviated names of people, if they are authors, if not, please mention them in acknowledgment: “The design and facilitation of the participatory process was conducted by a research team that included an expert on antimicrobial issues and pig and poultry farming and a person trained in participatory approaches and the ImpresS ex-ante method. A third researcher was in charge of checking the correct application of this method. Four researchers also observed the participatory workshops to later analyze the process from a sociological perspective.”
As a recommendation, in the method section, please provide a list of the main questions prepared at the beginning of the meeting.
Also present the limitations of the study, especially in the context of holding a meeting.
State your proposed solutions for holding such a meeting in other European countries as a suggestion in the final paragraph.
Author Response
- The result dedicated in the abstract is very general, for example, what tools can “standardized monitoring tools” be? Any recommendation?
- The conclusion part doesn’t reflect the result. It is only a suggestion. In the end, what solution was decided? Furthermore, please note that the complexity of sentences is high and long sentences will tire the reader (in the abstract).
The initial conclusion states that « Strong government support might help to boost initiatives and lift the lock-in effects resulting from economic competition between animal sector operators. It would be desirable for these issues to be addressed in the coming update of the French Plan Ecoantibio, and to involve other actors of the food chain »
Following the comment, we made it more active, as it is in reality, and wrote (Line 403) :
“Strong government support might help to boost initiatives and lift the lock-in effects resulting from economic competition between animal sector operators. The opportunity of the coming update of the French Plan Ecoantibio should allow to push actions on the standardization and spreading of monitoring tools at the farm level, and to work on communication towards the citizens.”
- Obviously, most of the stated results have already existed in the minds of many veterinarians and other parts of the related industries. State the main solution for coordination between different organizations and teams. Novelty is not well presented.
We agree that probably the ideas presented in the results were already present in the mind of many veterinarians, as the reviewer comments. The real novelty is that various people having diverging interests in the production chain were able to agree to prioritize these ideas as major components to improve the use of antimicrobials in the pig and poultry industries. Furthermore, the people involved acted as representatives of these production sector in the French industry, which gives more impact to the results. One of the consequences is that the authorities, that had a representative in the group, will impulse actions to implement these ideas. Also, these strong positions will allow to initiate concrete actions within the living lab.
We added on Line 392 : “The ideas issued from the group were probably present in the mind of many veterinarians; the real novelty is that various stakeholders having diverging interests in the production chain were able to agree to prioritize these items as major components to improve the use of antimicrobials in the pig and poultry sectors”.
- The keyword is good but not well. The items can improve using Mesh term. My recommendation: Please delete pig; poultry. Add “drug managing”; and “monitoring tools”; “antibiotic-free meat”.
We agreed and made the changes
- Related to the following sentence, it would be ideal for the authors to give some suggestions for Diagnostic, Control, and Monitoring tools to help future industries and veterinarians distinguish the best antibiotics: “In 2031, in France, the proper use of antimicrobials in the poultry and pork industries focuses on "better" and not just "less" antimicrobial use”.
We cannot provide these elements because the participatory process did not go in details on that point. It will be a possible further step of the living lab to work on these issues.
- Here, please give abbreviated names of people, if they are authors, if not, please mention them in acknowledgment: “The design and facilitation of the participatory process was conducted by a research team that included an expert on antimicrobial issues and pig and poultry farming and a person trained in participatory approaches and the ImpresS ex-ante method. A third researcher was in charge of checking the correct application of this method. Four researchers also observed the participatory workshops to later analyze the process from a sociological perspective.”
There is already a sentence in the acknowledgments (Line 428) :” The authors thank Florence Beaugrand, Nikky Millar and Marie-Hélène Pinard-Van-der-Laan, who played the role of observers of the process during the meetings ». For the other points, we added the initials of the authors involved on Lines 357-359, as proposed.
As a recommendation, in the method section, please provide a list of the main questions prepared at the beginning of the meeting.
We did not have a list of questions, but we introduced the goals of the meeting and made people work on the issue using active involvement. The goals of each meeting were designed based on the situation reached during the previous meeting.
Also present the limitations of the study, especially in the context of holding a meeting.
We added on Line 177 : “The size of the group proved to be relevant to be able to handle the participatory approach that is well designed to facilitate the involvement of the participants “
State your proposed solutions for holding such a meeting in other European countries as a suggestion in the final paragraph.
We added at the end of the final paragraph (Line 409) : “The participatory approach proved to be a fruitful option to to bring stakeholders with divergent interests to discuss on a unifying subject.